# Laboratory Investigation on the Shrinkage Cracking of Waste Fiber-Reinforced Recycled Aggregate Concrete

**DOI:** 10.3390/ma12081196

**Published:** 2019-04-12

**Authors:** Xiaoxin Wu, Jinghai Zhou, Tianbei Kang, Fengchi Wang, Xiangqun Ding, Shanshan Wang

**Affiliations:** 1School of Civil Engineering, Shenyang Jianzhu University, Shenyang 110168, China; wuxiaoxin@sjzu.edu.cn (X.W.); Kangtianbei@stu.sjzu.edu.cn (T.K.); cefcwang@sjzu.edu.cn (F.W.); 2School of Material Science and Engineering, Shenyang Jianzhu University, Shenyang 110168, China; Xiangqunding@126.com or dingxiangqun@sjzu.edu.cn; 3School of Foreign Languages, Shenyang Jianzhu University, Shenyang 110168, China; wangshanshan@stu.sjzu.edu.cn

**Keywords:** recycled aggregate concrete (RAC), waste fiber reinforced, shrinkage cracking, X-ray industrial computed tomography (ICT), mechanical properties

## Abstract

This paper aims to study the effectiveness of adding waste polypropylene fibers into recycled aggregate concrete (RAC) on shrinkage cracking. The influences of fiber properties (length and content) on the shrinkage performance of RAC are investigated. Firstly, through the plat-ring-type shrinkage test and free shrinkage test, both of the early age and long-term shrinkage performance of waste fiber recycled concrete (WFRC) were measured. Then, X-ray industrial computed tomography (ICT) was carried out to reflect the internal porosity changes of RAC with different lengths and contents of fibers. Furthermore, the compressive strength and flexural strength tests are conducted to evaluate the mechanical performance. The test results indicated that the addition of waste fibers played an important role in improving the crack resistance performance of the investigated RAC specimens as well as controlling their shrinkage behaviour. The initial cracking time, amount and width of cracks and shrinkage rate of fiber-reinforced specimens were better than those of the non-fiber-reinforced specimen. The addition of waste fibers at a small volume fraction in recycled concrete had not obviously changed the porosity, but it changed the law of pore size distribution. Meanwhile, the addition of waste fibers had no significant effect on the compressive strength of RAC, but it enhanced the flexural strength by 43%. The specimens reinforced by 19-mm and 0.12% (volume fraction) waste fibers had the optimal performance of cracking resistance.

## 1. Introduction

The quantity of construction and demolition waste (C&DW) is being produced annually around the world [1], which aroused environment pollution and resource depletion. All over Europe, about 320 to 380 million tons of C&DW are generated every year [2]. By contrast, in China, the amount of C&DW is approximately 640 million every year, with an average increasing rate of 8% per year [3,4]. The replacement of natural aggregates (NAs) with recycled aggregates (RAs) can decrease the amount of concrete waste that otherwise would be disposed in landfills. Moreover, it can relieve the requirement of the construction industry on new NAs [5]. Therefore, the use of RAs is essential for realizing a sustainable construction industry, remarkable in environmental preservation, and meaningful in improving resource efficiency [6,7,8]. There have been a great number of studies on the recycling of aggregates in the structural concrete manufacture.

Ample researches reported that the old mortar particles with relatively high porosity and low strength [9,10,11,12,13,14,15,16] lead to the inferior properties of RAs compared with NAs. Among those properties, shrinkage is the most affected factor, which causes a great weakness in the use of this type of material. Pedro et al. [17] reported that the mechanical performance, durability and shrinkage of the recycled aggregate concrete (RAC) are affected by RAs with different sources. They found that concrete containing the recycled coarse aggregate without mortar has superior mechanical performance compared to the natural aggregate concrete (NAC). However, the shrinkage resistance of recycled coarse aggregate concrete was less than this of natural coarse aggregate concrete. Manzi et al. [18] researched that short and long-term behaviour of structural concrete with RAs by adjusting and selecting the content and grain size distribution of concrete waste with a high content of RAs (ranging from 27% to 63.5% of total amount of aggregates). They conclude that shrinkage and creep, combined with porosity measurements and mechanical investigations, are fundamental features to assess structural concrete behaviour. Vegas et al. [19] studied shrinkage of RAs at 1250 h and concluded that the shrinkage value of 0.07 mm/m, more than triple the value obtained in concrete with NAs (0.02 mm/m). Fernández et al. [20] tested at 203 days and concluded that concrete types with RAC received more than double shrinkage compared to concrete with NAs. Ismail and Ramli [21] studied the strength and drying shrinkage performances of treated coarse RAC. Although the particle density, water absorption, mechanical strength remarkably and drying shrinkage concrete of RAs are worse than those of NAs, the effect of the combination of these two surface treatment methods improved those performances after treated, almost the same to those of NAs. The large drying shrinkage of RAC inhibits strength development and increases cracks. The shortcomings of RAC can be limited by adding reinforcing fibers, and crack propagation can be controlled [22,23,24].

There are varieties of fibers can be mixed in the RAC, such as polymeric fibers, steel fibers, natural fibers, etc. [25]. Akça et al. [26] studied the usability of polypropylene fiber in RAC to be used primarily in field concrete different combinations generated using the RAs and polypropylene fiber. Different polypropylene fiber contents have been introduced into concrete types that have a different amount of RAC, such as 0, 1% and 1.5% by volume. The result concluded that types of aggregate affected concrete compressive strength and fiber content affected flexural and splitting tensile strength besides aggregate type. Nam et al. [27] investigated the effects of polyvinyl alcohol fibers and nylon reinforced fibers on the mechanical properties and shrinkage cracking of recycled fine aggregate concrete. The results showed addition of fibers at a small volume fraction in RAC was more effective for drying shrinkage cracks than for improving mechanical performance. Kim et al. [28] researched the effects of recycled PET fiber reinforcement on shrinkage cracking of cement-based composites through pullout test with a different fraction of volume from 0.1% to 1.00%. The result indicated that the fraction of fiber volume affected plastic shrinkage cracking and the embossed type fiber which had advanced mechanical bond strength also possessed the best resistance to plastic shrinkage cracking. Skarzynski and Suchorzewski [29] studied the mechanical and fracture performances of concrete reinforced with recycled and industrial steel fibers. Through wedge splitting test and the analyses of the 3D cracking phenomenon with both recycled steel fibers and industrial steel fibers, it concluded that the mechanical performance could be improved with steel fibers and industrial steel fibers and cracking areas could be restrained by concrete with steel fibers and industrial steel fibers. Katkhuda and Shatarat [30] studied the improvement of the mechanical performances of RAs produced by adding the different volume of chopped basalt fibers (BF), including 0.1%, 0.3%, 0.5%, 1%, and 1.5%. The result showed that using chopped BF minimally enhanced the compressive strength of the concrete, but significantly improved its flexural and splitting tensile strengths. Moreover, the optimum BF content that produced the same splitting tensile and compressive strength as NAs was 0.5% for untreated RAs and 0.3% for treated RAs, while the flexural strength was 0.3% for untreated RAC and 0.1% for treated RAC. Although studies on the effects of using reinforcing fibers in RAs have attracted enough attention, the study about using of waste fibers in RAC is very few.

From the environmental viewpoint, adding the waste fibers into RAC is promising to improve its performance and to reduce potential pollution sources. As a combination type of green concrete, it can be reused for waste fibers and construction waste. Waste fiber recycled concrete (WFRC) not only promotes the application of recycled concrete but also is beneficial for environmental protection. Furthermore, to take economic efficiency into consideration, the utilization of waste fibers in structural concrete manufactures is meaningful to minimize the costs of fibers. Thus, it is necessary to investigate the usability of the waste fibers with RAC, especially its influence on shrinkage performance.

The objective of this study is to investigate the effectiveness of adding waste polypropylene fibers into RAC. Firstly, the plat-ring-type shrinkage test is conducted to study the shrinkage cracking performance under the restrained conditions. Meanwhile, the free shrinkage test is used to measure the rate of shrinkage in the unconstrained state. Then, the X-ray industrial computed tomography (ICT) is carried out to reflect the internal porosity of RAC. Additionally, the compressive strength and flexural strength were tested to evaluate the mechanical performance.

## 2. Materials and Methods

### 2.1. Materials

#### 2.1.1. Recycled Coarse Aggregates

In this experiment, the coarse aggregates were recycled from waste C40 concrete, which was crushed and cleaned manually. In order to remove oversize aggregates, the recycled coarse aggregates needed to be sieved before mixing. The maximum size of recycled coarse aggregates was 25 mm. Finally, the recycled coarse aggregates with a particle size of 5 to 25 mm were formed, which can be seen in Figure 1.

#### 2.1.2. Fine Aggregates

In this experiment, all the fine aggregates were natural river sand produced in the Hun River. The fineness modulus of the fine aggregates is 2.7; the moisture content is 4.15%; the density is 2610 Kg/m^3^. The gradation curves for coarse and fine aggregates are showed in Figure 2.

#### 2.1.3. Waste Fibers

End-of-life carpets (ELCs) were made into fiber segments with lengths of 12 mm and 19 mm by removing impurities, splitting and shearing. The chemical composition is polypropylene. Figure 3 showed the treating process of the fibers.

Table 1 provides the physical and mechanical properties of single textile waste polypropylene fiber. The Coefficient of Variation (CV) of the polypropylene waste fiber mechanical property is slightly higher than that of new polypropylene fiber, but the difference is not very significant.

#### 2.1.4. Cement

The cement used in this experiment was ordinary Portland cement labelled P.O.42.5. The properties of compositions of ordinary Portland cement are given in Table 2, Table 3 and Table 4.

### 2.2. Specimens

#### 2.2.1. Concrete Mixes

There are some differences in mixture design between the RAC and natural aggregates concrete. Due to the existence of old cement mortar attaching on the surface of recycled coarse aggregates, the recycled aggregates contain more voids than nature aggregates. Meanwhile, the RAC needs relatively more water to obtain the same workability as the natural aggregates concrete. According to DG/TJ 08-2018-2007 [31], water consumption can be divided into two parts: basic water consumption for mixing ordinary concrete and additional water consumption for recycled concrete. For the RAC with 100% replacement ratio, the additional water consumption is suggested to be 20 kg/m^3^. The waste polypropylene fibers do not absorb water. Thus, the influence of the waste fiber on the moisture content in the mixture ratio can be ignored.

The RAC specimens were divided into seven groups named FRC01 to FRC07, which contains different length and volume fraction of waste fibers (Table 5). Moreover, the NAC specimens were produced as a control group marked as NC. Before the mixing process, water absorption of recycled coarse aggregates was tested to be 2.125%. The mixture proportions of RAC were designed after trial mixing and adjustment for several times. Then, the water-cement ratio was set as 0.5. Consequently, the sand ratio was 38%. Additionally, to make the RAC meet the requirements of the slump and workability, referring to the standard [31], an additional 20 kg/m^3^ of water was added to the RAC. After that, the slump of RAC was tested to be the same as that of NAC. The mixture proportions of concrete specimens are listed in Table 5.

#### 2.2.2. Specimens for Plat-Ring-Type Test

The testing devices of the plat-ring-type test consisted of the baseboard, inner and outer steel ring, and the concrete specimen which was concreting between the inner and outer steel ring. After curing for 24 h, the outer steel ring was removed to form the concrete ring specimen restrained by the steel ring. The size of the experimental facility includes that the inner diameter of the inner steel ring is 281 mm, the outer diameter is 305 mm, and the height is 100 mm. The size of concrete ring is that the inner diameter is 305 mm, the outer diameter is 425 mm, and the height is 100 mm, as detailed in Figure 4. The prepared ring-shaped specimens can be seen from Figure 5.

#### 2.2.3. Specimens for Free Shrinkage Test

For the measurement of the concrete free shrinkage, different methods have been used in different studies. Suitable devices for testing the free shrinkage of RAC were assembled referring to other studies. The specimens were 500 mm × 100 mm × 100 mm prisms. The testing equipment and specimens are shown in Figure 6 and Figure 7, respectively.

### 2.3. Testing Methods

#### 2.3.1. Plat-Ring-Type Test

In order to study the shrinkage cracking behaviour of concrete specimens, plat-ring-type and free shrinkage test were employed. The testing instruments are shown in Figure 8 and Figure 9, respectively.

The plat-ring-type test was conducted by the ASTM C1581 [32]. Because of the restriction of the inner steel ring to the outer concrete ring, the concrete outer ring cannot shrink freely, which leads to the cracking of the concrete ring due to the tensile stress in the direction of the ring. Therefore, researchers can not only evaluate the cracking performance of the concrete by observing the number of cracks, the length of cracks, and the accumulative and maximum width of cracks with the change of time directly, but also can calculate strength and modulus of elasticity by studying the stress-strain curve of specimens with the strain gauges fixed to the outer concrete ring and inner steel ring. The specimens had a thickness of 100 + 5 mm and width of 125 + 2 mm.

In this experiment, in the middle of the inner steel ring of the specimen was bound to a group of strain gauges, and one group of strain gauges includes four strain gauges which are at an angle of 90 degree while the outer concrete ring was bounded to two strain gauges (Figure 5). Firstly, the strain gauges were connected to the strain-stress testing device, and the data was collected after the switch on and adjusting the balance. Secondly, the strain gauges which were bounded to the outer concrete ring and the inner steel ring were connected to the static strain testing device (Figure 8a). Finally, collecting the data at every stage of measured the age of concrete and the change situation of the concrete strain-stress can obtain by calculating the strain-stress of the steel ring.

To observe the cracking phenomenon, the specimens were placed into the constant temperature and humidity chamber at a temperature of 20 ± 2 °C and a relative humidity of 60 ± 5%. Researchers observed the outer surface of specimens to find whether cracks were appearing, and recorded the time of initial cracking. After the concrete cracks appeared, researchers used the crack test instrument (Figure 8b) to measure the width and height of cracks. The crack testing instrument can display the original appearance of cracks of the measured object on the colour screen and measure the length of core cracks using a vernier caliper. However, if the crack is deflected, the sum of the lengths of each crack is taken as the length of this crack, and the sum of the maximum widths of all cracks appearing on each specimen is taken as the cumulative width of cracks. In the first seven days, data were recorded every 6 h. From the 8th day to the 150th day, data were recorded every 24 h.

As can be seen in Figure 8, the strain collection used a static strain gauge, called XL2101B3+ strain gauge, which was produced by Xieli Science and Technology Co. Ltd. It has 16 measuring points and can be scanned and recorded automatically. The PTE-E40 comprehensive cracking tester, produced by Wuhan Botest Instrument Co. Ltd., (Wuhan, China) was applied in cracks measurement. The comprehensive cracking tester integrates the micro-image processing and ultrasonic testing technology, which enables the equipment to measure the width and depth of fractures at the same time.

#### 2.3.2. Free Shrinkage Test

The free shrinkage test was devised according to the ASTM C157 [33] and GB/T 29417-2012 [34]. Firstly, the steel mould was made to the dimension of size in Figure 6. Then, the Polytetrafluoroethylene (PTFE) boards were placed between the steel mould and the specimen, in order to reduce the friction resistance substantially. Thus, the concrete specimens would shrink under free condition. Moreover, the micrometres were fixed on the centre of the steel probes, which was embedded in both ends of the specimen. Finally, the internal temperature could be determined by the digital temperature tester and thermocouple (Figure 9a). To measure the behaviour of free shrinkage in a constant condition, the RAC specimens were placed into chambers at a temperature of 20 ± 2 °C and a relative humidity of 60 ± 5%. In the first 24 h, data were recorded every 2 h. From the 24th hour to the 72nd hour, data should be recorded every 6 h. After the 72nd hour, the data of micrometre were recorded every 24 h during all stage of concrete setting. The setup of the free shrinkage test can be seen in Figure 9b.

#### 2.3.3. X-Ray Computed Tomography

The X-ray ICT scanning system was conducted to scan the RAC specimens at different curing periods. The internal structure of each test specimen was imaged using the TomoScope HV ICT system produced by Werth Inc. (Hannover, Germany). It equipped with a microfocus X-ray tube capable of attaining a resolution of 4.5 μm. The applied voltage and power were 225 KV and 280 W, respectively. The working turntable for reconstructed specimens placing and schematic diagram of the ICT scanning system can be seen from Figure 10. The ICT scan tests were performed on cylinder specimens with a diameter of 10 cm and a height of 20 cm. The specimens were cured in a room with a relative humidity of 60 ± 5% and temperature of 20 ± 2 °C.

According to the scanning theory of X-ray ICT, the air void in the concrete specimen was defined as a defect. The defect could be recognized and imaged utilizing collecting and processing the density difference between air voids and aggregates or cement mortars. Thus, two-dimensional and three-dimensional images, as well as the three-dimensional parameters of reconstructed specimens, can be obtained by VG Studio Max 3.0.2. The images can be seen in Figure 11.

#### 2.3.4. Compressive Strength and Flexural Strength Tests

Compressive strength and flexural strength were conducted in order to determine mechanical properties of WFRC. The flexural strength test was carried out with the same prismatic specimens (100 × 100 × 500 mm) after the free shrinkage tests in accordance with GB/T 50081 [34]. Then the fractured prismatic specimens were cut into double amounts of cubic specimens (100 × 100 × 100 mm) at both edges of the prismatic specimens. The compressive strength tests were performed using the cubic specimens in accordance with GB/T 50081 [35].

## 3. Results and Discussions

### 3.1. Shrinkage Cracking Performance

The concrete produced stress when the concrete shrunk restrained by the inner steel ring, and the stress increased gradually with the development of the time. However, the outer concrete would have cracks when the stress of the inner concrete over the compressive strength at the corresponding age. The plat-ring-type test based on the cracking caused by shrinkage of concrete under constraint conditions analyzes the influence of waste fibers on the cracking performance of WFRC qualitatively and quantitatively. The shrinkage cracking performance of each ring concrete specimen is displayed in Figure 12.

As presented in Figure 13, the order of the initial cracking time of each specimen is FRC01, FRC02, FRC04, FRC07, FRC03, FRC05, and FRC06. The initial cracking time of the FRC01 was the earliest, at the 79th hour. When compared to the FRC01, the initial cracking time of FRC02, FRC03 and FRC05 were 34 h, 89 h and 91.5 h later, respectively. Besides, the initial cracking time of specimens with 19-mm waste fibers was later than that of the specimens with 12 mm waste fibers and the initial cracking time of the specimen with 0.12% volume fraction was later than that of specimens with other volume fraction. The initial time of the concrete with 19-mm waste fibers and 0.12% volume fraction is the latest, about 3.7 days delayed. Therefore, it can be concluded that the added waste fibers affected the initial cracking time effectively.

The strain at the initial cracking of the concrete and stress of the steel ring before cracking can be seen in Figure 13. As illustrated, with the growth of time, the outward force of the circular concrete specimen was also gradually increasing. Up until the cracks appeared, the stress would decrease and release. The tensile stresses in specimens were lower than the tensile strength before concretes cracked. The initial cracking time of FRC06 was the latest, and the stress was 2.0 MPa which was lower than the average value. FRC04 and FRC07 cracked almost at the same time, and the stresses were 1.79 MPa and 1.8 MPa, respectively. While, the cracking strain of FRC04 was only 2383 με, much higher than that of FRC07, 1682 με. Therefore, with a 0.16% volume fraction, the crack resistance performance of the specimens with 19-mm waste fibers was higher than that of the specimens with 12 mm.

Figure 14 plots the total amount of cracks and the penetrating cracks in concrete specimens at different ages. For the number of cracks, the total number of cracks in each specimen during the 75 d would no longer be changed, compared with the FRC01, the addition of waste fibers reduced the number of cracks before 75 d, effectively. While after 75 d, the growth in the number of cracks was not obvious. It can be inferred that the crack resistance effect was not only to eliminate or restrain the development of macro-cracks but to gradually refine the macro-cracks with larger width into micro-cracks or invisible cracks with eyes.

In this paper, the maximum crack width in the circular specimen of concrete at each age was determined as one important index to evaluate the crack resistance performance of WFRC. As shown in Figure 15, the maximum width of cracking developed rapidly in the first 90 days and tended to be stable after 90 days. The maximum width of cracking of the concrete without any fiber on the initial cracking was 0.07 mm, and the maximum width of cracking was 0.43 mm, 0.48 mm, 0.49 mm at 28 d, 90 d and 150 d respectively, which was all larger than that of WFRC. The maximum widths of cracking of concrete with 12 mm waste fibers and 0.08%, 0.12% and 0.16% volume fraction were 0.33 mm, 0.39 mm and 0.39 mm in 90 days, which decreased by 31.2%, 18.8% and 18.8%, respectively. The maximum widths of cracking of concrete were 0.36 mm, 0.41 mm and 0.44 mm at 150 d, which decreased by 28.6%, 16.3% and 10.2%, respectively. The maximum widths of cracking of concrete with 19-mm waste fibers and 0.08%, 0.12% and 0.16% volume fraction in 90 days decreased by 31.2%, 18.8% and 18.8%, respectively. The maximum widths of cracking of concrete were 0.42 mm, 0.39 mm and 0.45 mm at 150 d, which decreased by 12.5%, 18.8% and 8.2%. It can be concluded that if the 19-mm fibers were mixed, the optimal volume fraction is 0.12% for cracking resistance performance.

Figure 16 plots the cumulative width of all cracks on circular concrete specimens at different ages. As can been seen in Figure 15, the order of width of the maximum crack in concrete from large to small is as follows: FRC01, FRC07, FRC04, FRC05, FRC03, FRC06, and FRC02. While, it can be also seen in Figure 16, the order of width of cumulative width cracks in concrete from large to small is as follows: FRC01, FRC02, FRC07, FRC05, FRC04, FRC03, and FRC06. Comparing the orders of the two groups, the specimens without fibers (FRC01) have the worst conditions. The addition of fibers can reduce both of the maximum width and cumulative width cracks. Moreover, the orders of the two groups were similar except for FRC02.

Compared with the non-fiber-reinforced group, the widths of cracking of specimens with 12 mm waste fibers and 0.08%, 0.12% and 0.16% volume fraction in 90 days were decreased by 6.8%, 22.3% and 10.1%, respectively. The specimens with 19-mm waste fibers and 0.08%, 0.12% and 0.16% volume fraction at 90 d decreased by 15.5%, 15.5% and 2.7%, respectively; and at 150 d by 14.7%, 20.9% and 3.4%, respectively. Therefore, the RAC reinforced by 0.12% volume fraction waste fibers has the optimal cracking resistance performance.

### 3.2. Free Shrinkage Behavior

The free shrinkage test was carried out to analyse the shrinkage of WFRC under unrestrained conditions, including early shrinkage (first 72 h) and long-term shrinkage (150 d). The early shrinkage includes the autogenous shrinkage (hydration reaction) and temperature shrinkage while the long-term shrinkage includes drying shrinkage and autogenous shrinkage.

The early shrinkage and long-term shrinkage of different kinds of WFRC were studied, making two specimens of each kind of WFRC and calculating the average amount of the two values. Figure 17 showed that the results of the early free shrinkage behaviour of WFRC. The test was carried out at a temperature of 20 ± 2 °C and a relative humidity of 60 ± 5% in a constant temperature and humidity chamber lasting 72 h.

It can be seen in Figure 17 that the shrinkage deformation of all specimens developed rapidly in the first 24 h, and the development speed gradually slowed down after 24 h. The shrinkage rate of FRC01 and FRC02 reached to 438 × 10^−6^ and 326 × 10^−6^ at 24 h, which accounted for 86.1% and 89.1% of the total of 72 h, respectively. The shrinkage rate of FRC01 was 508.7 × 10^−6^ at 72 h, while the shrinkage rate of FRC04 was 427 × 10^−6^, which was 16.07% lower than that of FRC01. Also, the FRC06 had the lowest shrinkage rate at 72 h, 30.41% lower than that of FRC01.

Figure 18 reveals the final results of the long-term free shrinkage behaviour of RAC. To ensure the accuracy of the test, the specimens placed in the chamber, keeping at a temperature of 20 ± 2 °C and a relative humidity of 60 ± 5% for 150 d.

Figure 18 illustrates that the 150-d shrinkage rate of RAC in the sequence is FRC01, FRC07, FRC04, FRC02, FRC03, FRC05 and FRC06, which is consistent with that of 72 h. It indicates that the early shrinkage rate of concrete is relative to the long-term shrinkage rate. Specifically, the shrinkage rate of FRC06 was the lowest, which was 30.2% and 9.72% lower than that of FRC01 in 10 d and 150 d, respectively. Also, the shrinkage rates of RAC reinforced by 19-mm waste fibers were lower than that with 12-mm fibers. Regardless of the long-term or early shrinkage performance, the shrinkage resistance effect of 19-mm fibers was better than 12 mm fibers. This is the same as the results of the plat-ring-type test.

### 3.3. Characteristics of Pore Structures

The X-ray ICT images were visually evaluated regarding the internal structures and porosity characters of the concrete specimens. The results of three-dimensional reconstruction and the images before and after porosity analysis were illustrated in Figure 19. It is inevitable that there are more or less pores in the cement concrete. The pores mainly exist in the cement mortar, including the air mixed during the process of stirring and vibrating, the voids left after evaporation of moisture, the defects in the raw materials, and the gap left by shrinkage distortion of the mixture. Front view of ICT images before and after porosity analysis can be seen in Figure 20.

As illustrated in Figure 21, the NAC specimen (labeled NC) had the lowest pore volume, 62,808 mm^3^, which was 36,693 mm^3^ less than that of the RAC specimen (FRC01). The addition of waste fibers had no significant effect on the porosity of recycled concrete. The differences of porosity among FRC01, FRC05, FRC06 and FRC07 were within ±0.6%. The specimens without fiber (NC and FRC01) contained all sizes of pores evenly from 0.01 mm^3^ to 200 mm^3^. In specimens reinforced by waste fibers (FRC05, FRC06, FRC07), the proportions of small pores (volume < 1 mm^3^) were significantly increased. The volume of micropores (volume < 0.1 mm^3^) in FRC07 and FRC06 were 34,463 mm^3^ and 23,922 mm^3^, accounting for 38.33% and 27.89% of all, respectively. Moreover, the volumes of small pores (volume < 1) in FRC07 and FRC06 were 62,397.16 and 55,498.88, which were greater than that of FRC01, 43,688.80 mm^3^. Nevertheless, the volume of micropores (volume < 0.1 mm^3^) in FRC01 was 10,523 mm^3^, only 10.58% of all, while the volume of big pores (volume > 10 mm^3^) was 26,404 mm^3^, which was larger than other specimens. From the above, compared to non-fiber-reinforced RAC, the proportions of small and medium pores of fiber-reinforced RAC were larger, on the contrary, the proportions of the big pores were much smaller. Thus, the uniformity of the pores was improved by adding waste fibers. Thus, it can be concluded that the addition of the fibers can improve the pore structures in concrete and the uniformity of the distribution of materials. These also can be regarded as an explanation of the phenomenon of the addition of waste fibers can reduce shrinkage and cracking of RAC.

### 3.4. Compressive Strength and Flexural Strength

Four replicate samples were tested to investigate the compressive strength of concrete. The results of compressive strength tests are shown in Table 6. For the non-fiber-reinforced concrete specimens, the mean value of compressive strength of natural aggregate concrete (marked as NC) was 44.52 MPa, which was higher than that of FRC01 by 11.6%. Additionally, in cases of fiber-reinforced recycled concrete, the relative strength of all specimens was lower than NC, from 7.35% up to 13.7%. These indicate that the compressive strength of concrete with 100% replacement ratio is lower than that of NAC, which consistent with the previous researches [9,13,22,36,37]. The values of the compressive strength of the fiber reinforced concrete were not found to be significantly different from the value of compressive strength of FRC01. In other words, the addition of fibers had no influence on the compressive strength of RAC specimens; the similar observations were previously reported by Xiao et al. [38] and Borg et al. [39]. The compressive strength of FRC06 is the highest relatively, and its porosity is lower than that of other specimens; the compressive strength of FRC07 is slightly higher than that of FRC04, and the porosity of FRC07 is lower than that of FRC04.So it can be inferred that the compressive strength of RAC is negatively affected by the porosity.

Table 6 lists the flexural strength for concrete specimens at 150 days. The mean value of flexural strength of FRC01 is 18.35% lower than that of NC specimens. The flexural strength of concrete specimens was slightly improved when a small volume fraction of waste fibers was used. The flexural strength of FRC02 is 5.58 MPa, which is close to that of non-fiber-reinforced NAC. The flexural strength increases with the increase in the fiber content, with a volume fraction of 0.12%, and the flexural strength of FRC03 and FRC06 increase by 13.1% and 14.1%, respectively. The maximum of flexural strength is turned to be FRC06, 7.12 MPa, reinforced by 19 mm and 0.12% (volume fraction) fibers.

## 4. Conclusions

This study investigated the effectiveness of adding waste polypropylene fibers into RAC. The shrinkage performance of WFRC was measured by Plat-ring-type shrinkage test and free shrinkage test. Then, X-ray ICT was carried out to reflect the internal porosity of RAC. Furthermore, the compressive strength and flexural strength were tested to evaluate mechanical performance. Based on the above analysis, the following conclusions could be drawn:(1)In the early age of RAC, the growth speed of shrinkage and cracking is fast, and the growth speed slows down with the increase of curing time. The number of cracks and shrinkage rate gradually stabilize after 90 days. The early shrinkage of all specimens is consistent with the long-term shrinkage.(2)The results of the plat-ring-type test indicate that the addition of the waste fibers cannot only delay the initial cracking time of the recycled concrete but also reduce the number of cracks in recycled concrete ring specimens and the width of cracks. As to the results of the free shrinkage test, the addition of the waste fibers can decrease the early and long-term shrinkage rate of the recycled concrete effectively.(3)The results of both the plat-ring-type shrinkage test and free shrinkage test show that the cracking resistance performance of specimens reinforced by 19-mm fibers is better than that of 12 mm. The RAC reinforced by 19 mm and 0.12% (volume fraction) waste fibers has the optimal performance of cracking resistance, especially.(4)The addition of a small volume of waste fibers will not change the porosity of recycled concrete, significantly, but it can change the law of pore size distribution. Still, the addition of fibers can make the pore size of recycled concrete more uniform.(5)This paper mainly focuses on the reinforcement of RAC by adding waste fibers. Although there is slightly higher variability in mechanical properties of fibers, the addition of waste fibers has a certain effectiveness on improving the performance of concrete. After adding the waste fibers, the flexural strength of RAC was improved. When the addition of the volume fraction of the 19-mm fiber was 0.12%, the flexural strength of recycled concrete was 43% higher than that of recycled concrete without fiber. The compressive strength of the concrete whose recycled fine aggregate replacement ratio was 100% was lower than that of NAC, and the addition of fiber has no significant effect on the compressive strength of recycled concrete specimens. The air void in concrete has a negative effect on compressive strength.(6)In this paper, only one concrete mix design is considered. More mix designs can be considered in further studies.

## Figures and Tables

**Figure 1 materials-12-01196-f001:**
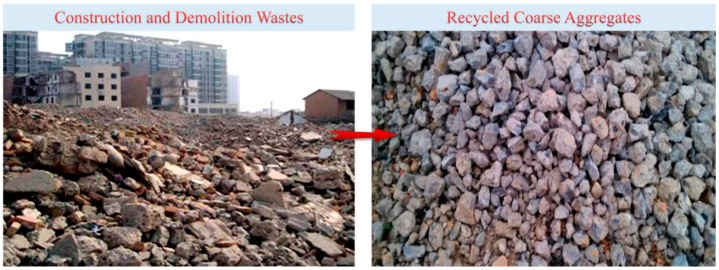
Recycled coarse aggregates.

**Figure 2 materials-12-01196-f002:**
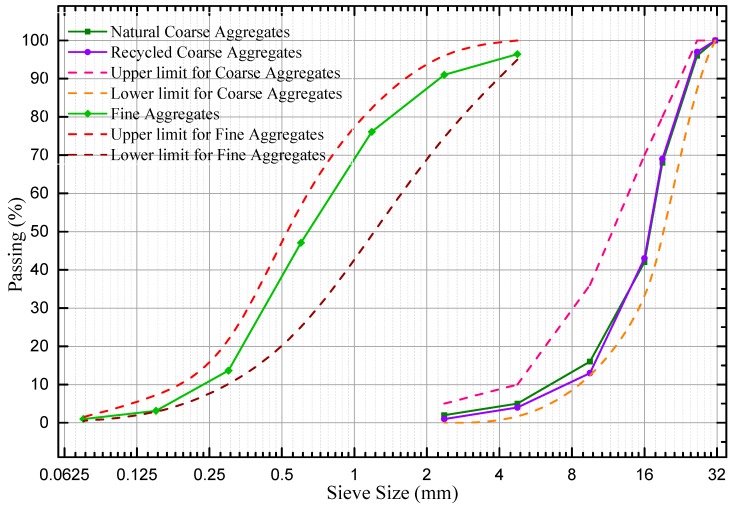
Aggregates gradating curves.

**Figure 3 materials-12-01196-f003:**
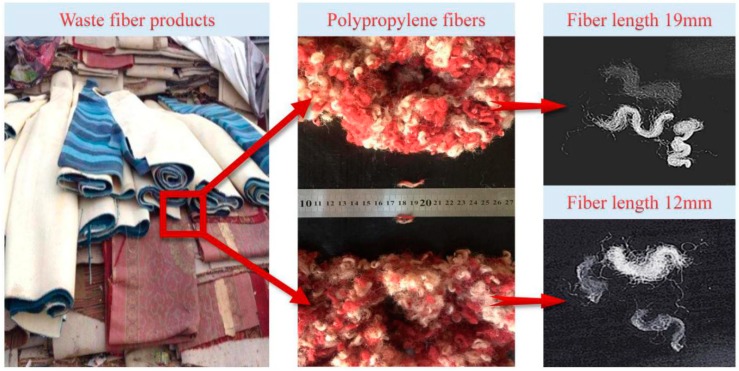
Waste polypropylene fibers.

**Figure 4 materials-12-01196-f004:**
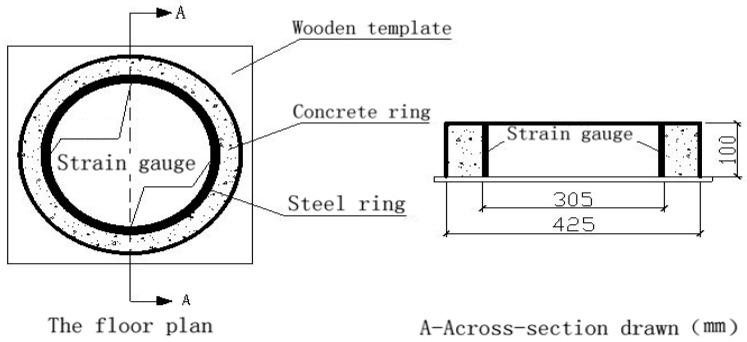
Schematic diagram of the plat-ring-type test.

**Figure 5 materials-12-01196-f005:**
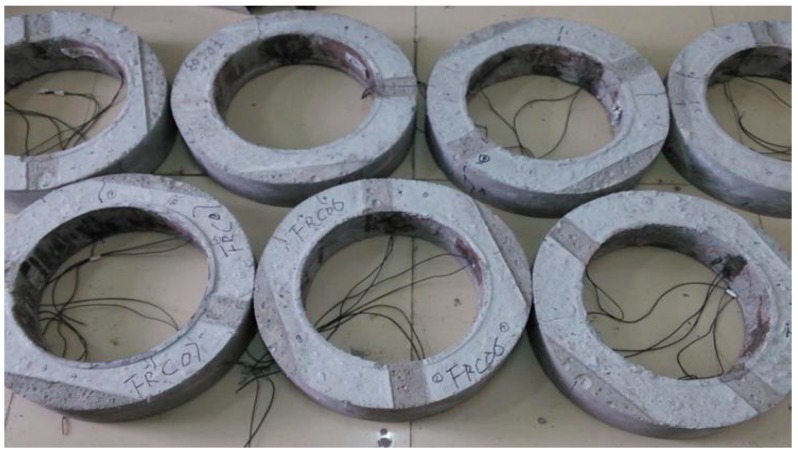
Specimens for the plat-ring-type test.

**Figure 6 materials-12-01196-f006:**
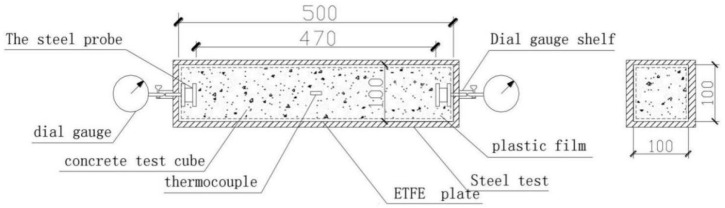
Schematic diagram of the free shrinkage test.

**Figure 7 materials-12-01196-f007:**
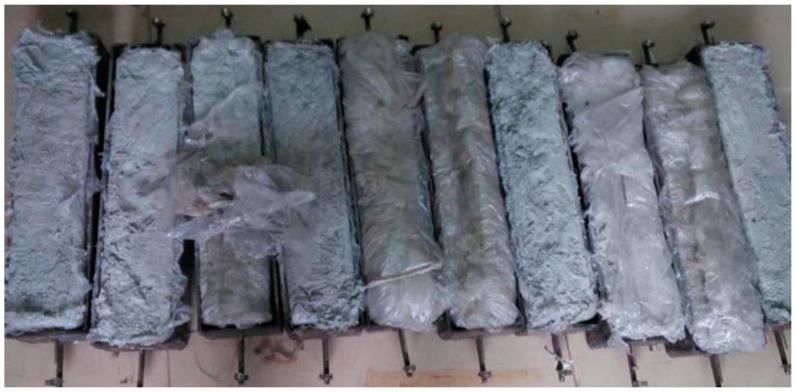
Specimens for the free shrinkage test.

**Figure 8 materials-12-01196-f008:**
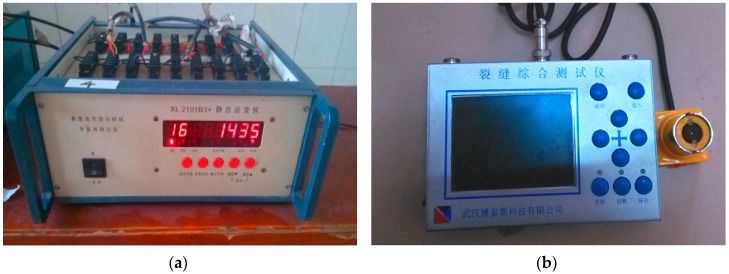
Instruments for the plat-ring-type test: (**a**) static strain testing device; (**b**) crack testing device.

**Figure 9 materials-12-01196-f009:**
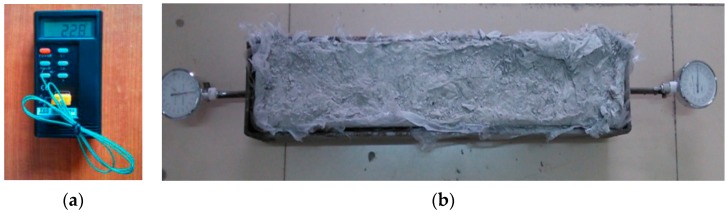
Instruments for the free shrinkage test: (**a**) Digital temperature tester and thermocouple; (**b**) Free shrinkage test setup.

**Figure 10 materials-12-01196-f010:**
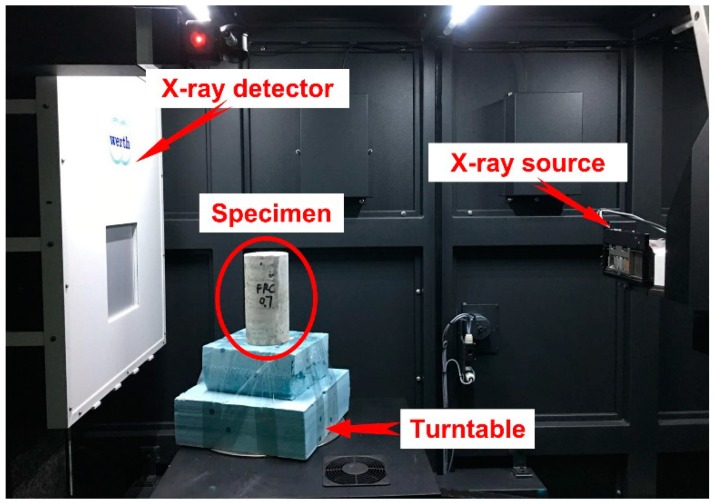
X-ray industrial computed tomography (ICT) scanning system.

**Figure 11 materials-12-01196-f011:**
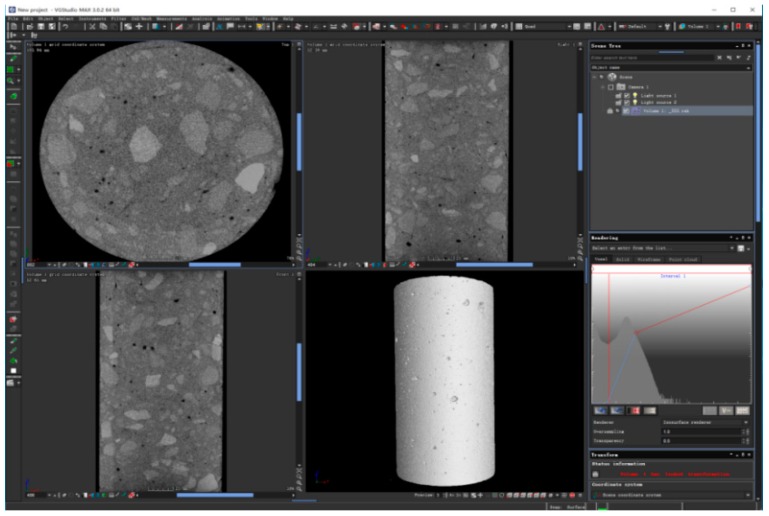
Screenshot of VG Studio Max 3.0.2 software.

**Figure 12 materials-12-01196-f012:**
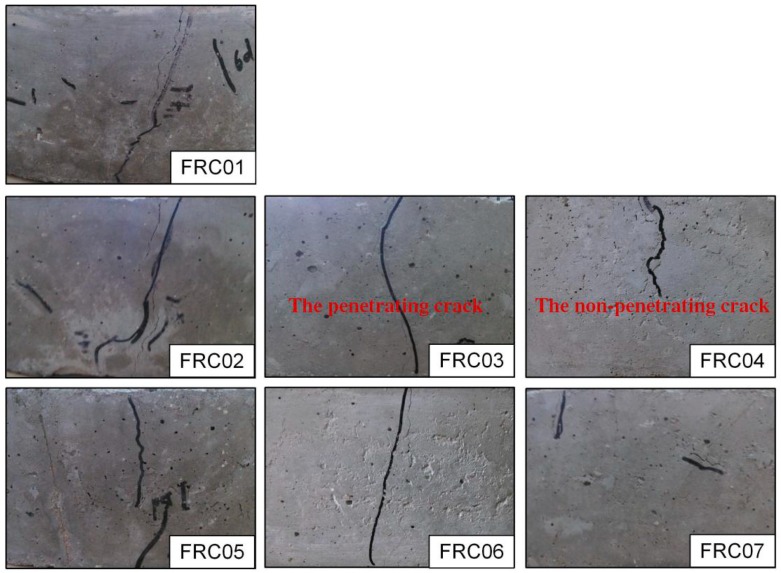
Cracking phenomena of circular concrete specimens.

**Figure 13 materials-12-01196-f013:**
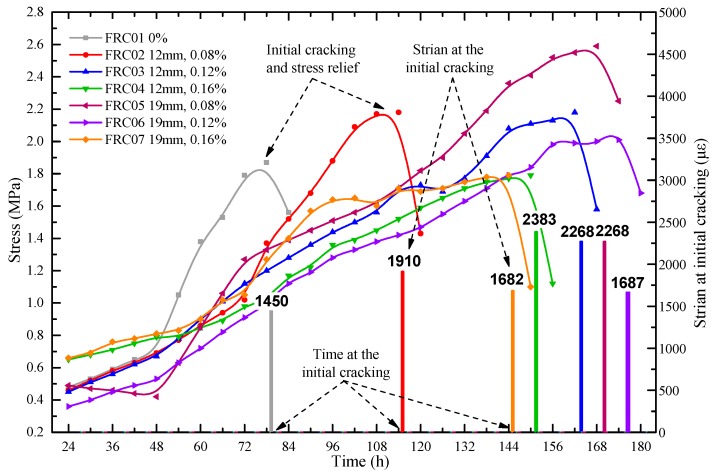
Time and strain at the initial cracking of the concrete specimens and stress of the steel ring before cracking.

**Figure 14 materials-12-01196-f014:**
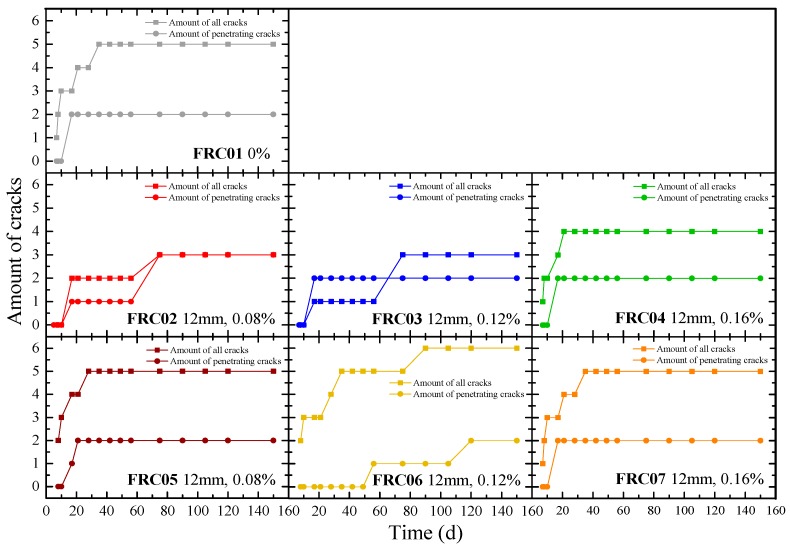
Amount of cracks in circular concrete specimens during 150 days.

**Figure 15 materials-12-01196-f015:**
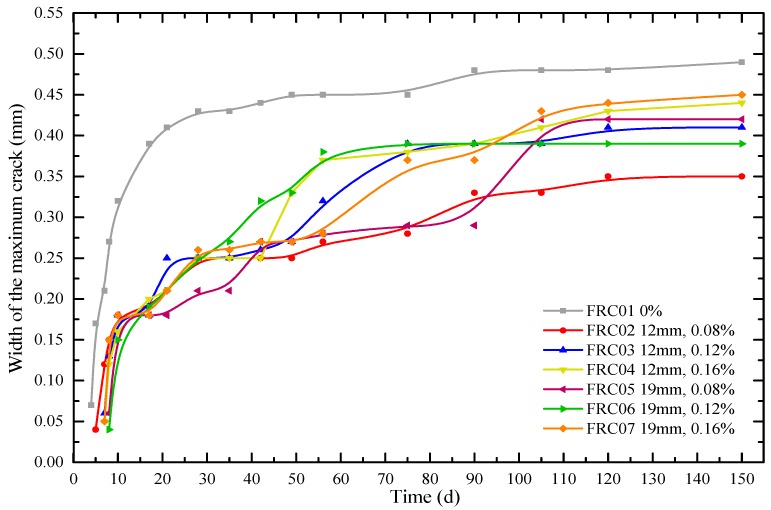
Width of the maximum crack in the concrete specimens during 150 days.

**Figure 16 materials-12-01196-f016:**
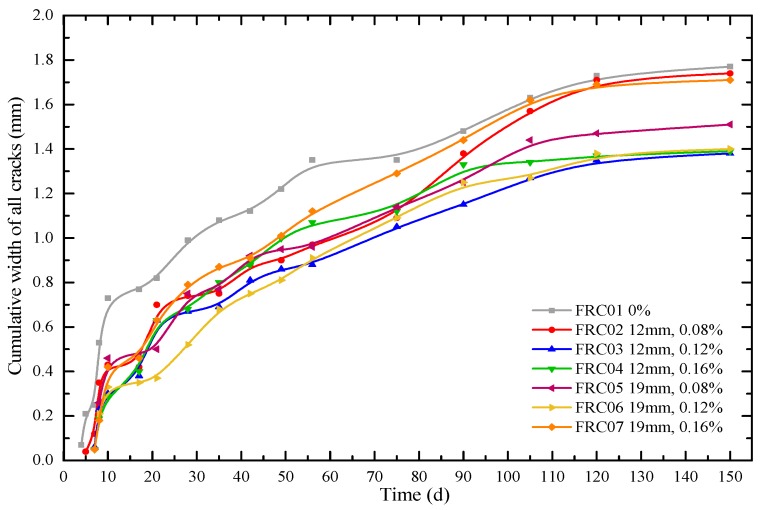
Cumulative width cracks in the concrete specimens during 150 days.

**Figure 17 materials-12-01196-f017:**
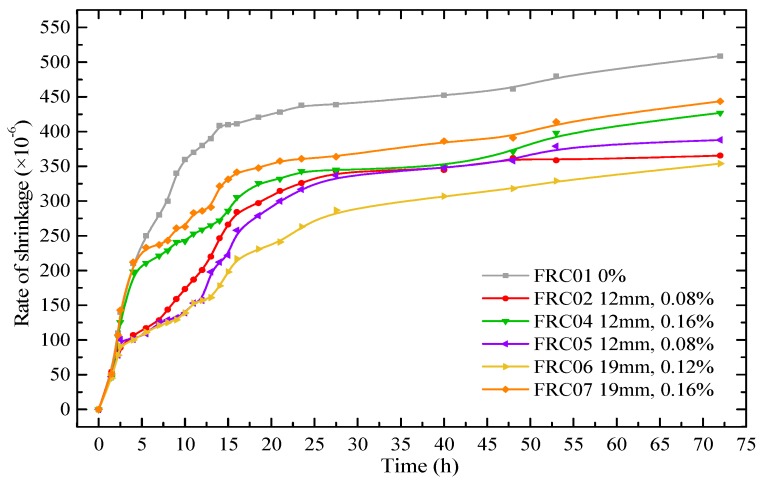
Free shrinkage deformation of prism concrete specimens during 72 h.

**Figure 18 materials-12-01196-f018:**
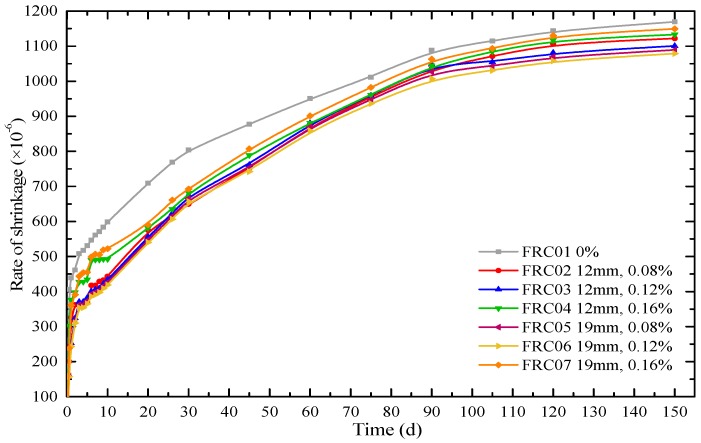
Free shrinkage deformation of the concrete specimens during 150 days.

**Figure 19 materials-12-01196-f019:**
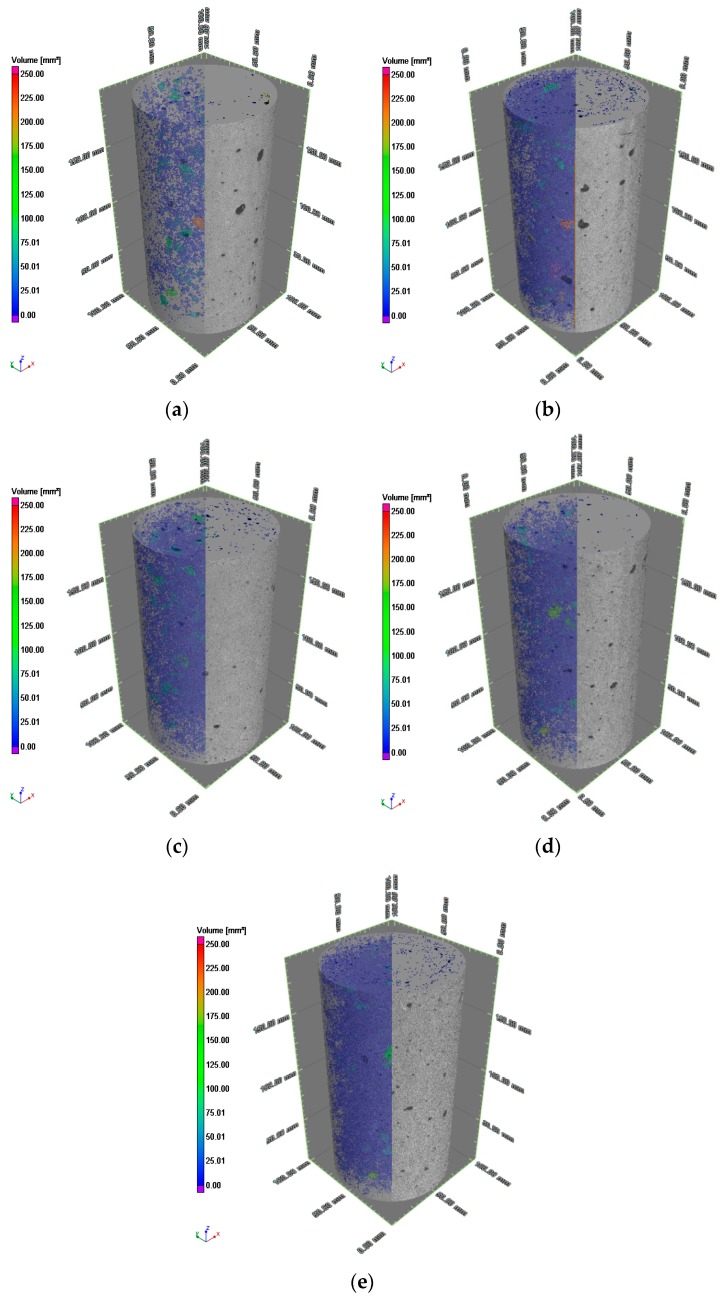
Three-dimensional reconstruction and porosity analysis images: (**a**) NC natural aggregates concrete; (**b**)FRC01 no fiber; **(c)** FRC05 with 0.08% fibers; (**d**) FRC06 with 0.12% fibers; (**e**) FRC07 with 0.16% fibers.

**Figure 20 materials-12-01196-f020:**
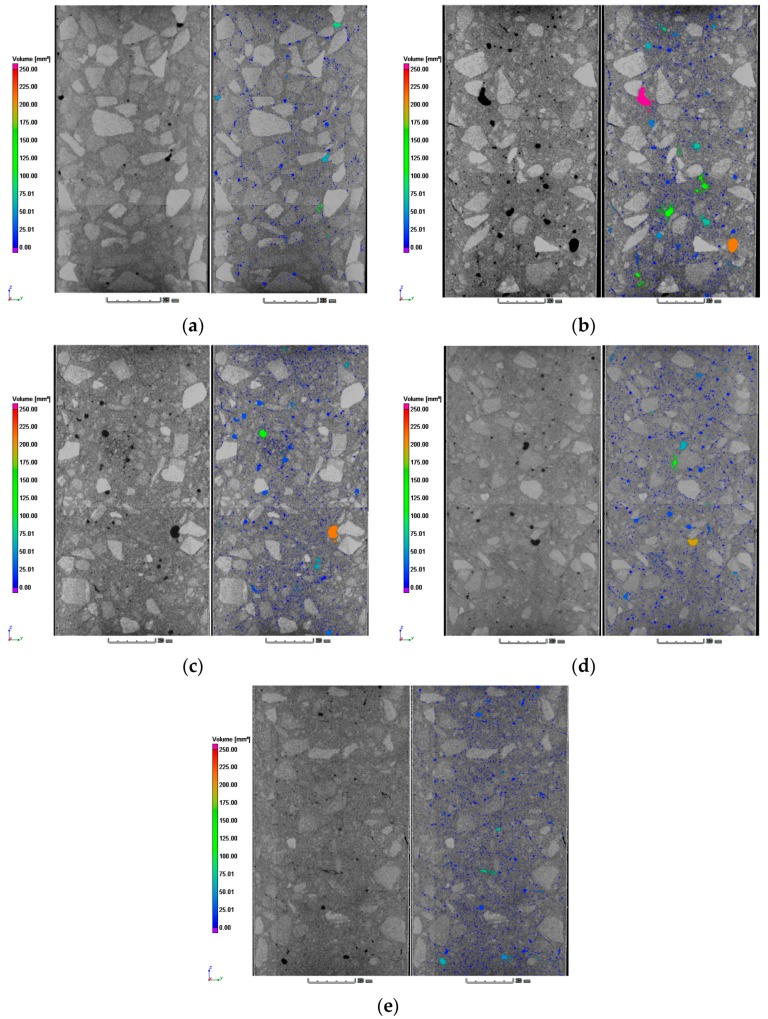
Front view of ICT images before and after porosity analysis: (**a**) NC; (**b**)FRC01 no fiber; (**c**) FRC05 with 0.08% fibers; (**d**) FRC06 with 0.12% fibers; (**e**) FRC07 with 0.16% fibers.

**Figure 21 materials-12-01196-f021:**
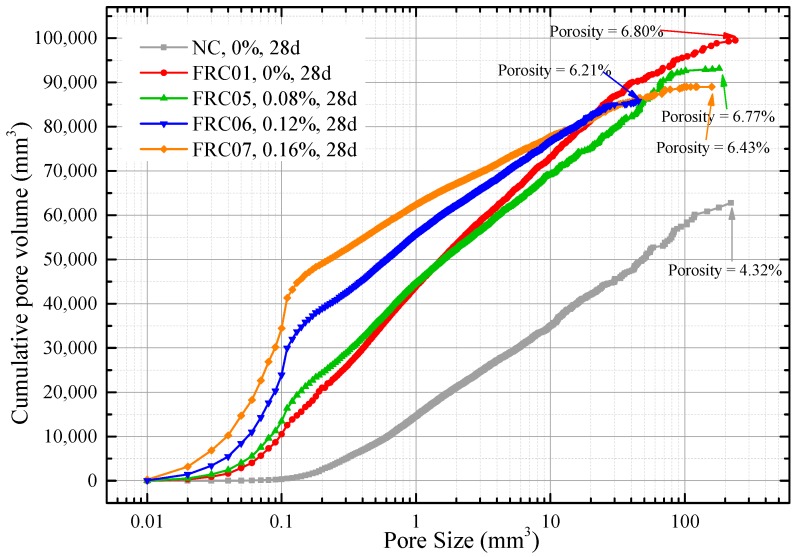
Pore volume of different sizes from ICT images porosity analysis.

**Table 1 materials-12-01196-t001:** Physical and mechanical properties of the waste polypropylene fiber.

Length (mm)	Density (g/cm^3^)		Water Absorption (%)		Fusion Point (℃)
12/19	0.91		<0.1		≈165
Elastic Modulus (MPa)		Breaking tensile strength (MPa)		Breaking Elongation (%)	
3.79 × 103		310 (CV = 16.5%)		18.5 (CV = 17.1%)	

**Table 2 materials-12-01196-t002:** Rate of cement clinker mineral composition.

Rate of Cement Clinker (%)			Mineral Composition (%)			
KH	SM	IM	C3S	C_2_S	C_3_A	C_4_AF
0.893	2.66	1.7	55.91	20.53	7.12	10.32

**Table 3 materials-12-01196-t003:** Chemical composition of cement.

Chemical Composition (%)				
SO_3_	MgO	Loss on Ignition	Insoluble Residue	Content of Alkali
2.49	3.54	1.0	0.55	0.51

**Table 4 materials-12-01196-t004:** Physical properties of cement.

SCA (m^3^/kg)	Normal Consistency (%)	Setting Time		Soundness	Compressive Strength (MPa)		Flexural Strength (MPa)	
		Initial Set	Final Set		3 days	28 days	3 days	28 days
348	25.0	2 h 20 min	3 h 50 min	Qualified	25.9	43.6	5.9	9.1

**Table 5 materials-12-01196-t005:** Mixture proportions of concrete specimens.

Group of Specimens	Water-Cement Ratio	Length of Fibers (mm)	Replacement Ratio of RAs	Volume Fraction of Fibers	Cement (kg/m^3^)	Sand (kg/m^3^)	RAC (kg/m^3^)	Water (kg/m^3^)
FRC01	0.55	0	100%	0	390	709	1156	215
FRC02	0.55	12	100%	0.08%	390	709	1156	215
FRC03	0.55	12	100%	0.12%	390	709	1156	215
FRC04	0.55	12	100%	0.16%	390	709	1156	215
FRC05	0.55	19	100%	0.08%	390	709	1156	215
FRC06	0.55	19	100%	0.12%	390	709	1156	215
FRC07	0.55	19	100%	0.16%	390	709	1156	215
NC	0.50	0	0	0	390	709	1156	195

**Table 6 materials-12-01196-t006:** Compressive strength and flexural strength of concrete specimens.

Group of Specimens	Compressive Strength (MPa)	Flexural Strength (MPa)
FRC01	39.82	4.96
FRC02	39.15	5.58
FRC03	40.12	6.31
FRC04	39.88	6.57
FRC05	39.95	6.24
FRC06	41.47	7.12
FRC07	40.89	6.95
NC	44.52	5.87

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
