# Peer review of "Laboratory Investigation on the Shrinkage Cracking of Waste Fiber-Reinforced Recycled Aggregate Concrete"

_materials, 2019, doi:10.3390/ma12081196_

Round 1
Reviewer 1 Report
- Add gradation curves for aggregates.
- Were there any admixtures added?
- Add a table for mechanical properties of fibers. It seems that the properties may vary. How can you make sure that the fibers used will have same properties?
- Provide information of devices, including the environmental chamber, used for testing
- It does not seem that there is a trend in initial cracking and amount of fiber waste added
- Please explain how amount of cracks increase then decrease
- Please add more discussion to Figure 14 etc. There is no trend. If you look at the results at 150 days, you will see that they are kind of random.
- Conclusions presented are ok, but this work has several uncertainties that need clarification. One of them is the properties of the fibers, which may vary. Also, this work is done based on one mix design.
Author Response
Thank you for your constructive comments on our manuscript entitled “Laboratory Investigation on the Shrinkage Cracking of Waste Fiber Reinforced Recycled Aggregate Concrete” (ID: materials-472621). All comments are very helpful for revising and improving our paper. We have studied comments carefully and have responded these questions point-by-point, which can be found in the response letter. All changes made to the original manuscript are highlighted in red.

Reviewer 2 Report
Comments in the pdf file

Author Response
Thank you for your constructive comments on our manuscript entitled “Laboratory Investigation on the Shrinkage Cracking of Waste Fiber Reinforced Recycled Aggregate Concrete” (ID: materials-472621). Those comments are all valuable and helpful for revising our paper. We have studied comments carefully and have made revisions point-by-point, which can be seen in the response letter. All changes made to the original manuscript are highlighted in red.

Round 2
Reviewer 1 Report
None
Reviewer 2 Report
"Point 13: Figure 18 and 19 - the descriptions are unreadable.
Response 13: These descriptions are about the dimensioning. If the figures are enlarged, they can be read. The editor could determine the size of these figures during the the typesetting process."
I know that the descriptions in figures 19 and 20 (now) are about the dimensioning and are not very important, but even enlarged are unreadable. This needs to be improved, or delete the descriptions. In general, the article now is ok. Good job.
This manuscript is a resubmission of an earlier submission. The following is a list of the peer review reports and author responses from that submission.